# *Mycobacterium tuberculosis* Deficient in PdtaS Cytosolic Histidine Kinase Displays Attenuated Growth and Affords Protective Efficacy against Aerosol *M. tuberculosis* Infection in Mice

**DOI:** 10.3390/vaccines12010050

**Published:** 2024-01-02

**Authors:** Kelly A. Prendergast, Gayathri Nagalingam, Nicholas P. West, James A. Triccas

**Affiliations:** 1Sydney Infectious Diseases Institute (Sydney ID), Faculty of Medicine and Health, The University of Sydney, Camperdown, NSW 2006, Australia; kelly.prendergast@sydney.edu.au (K.A.P.); gayathri.nagalingam@sydney.edu.au (G.N.); 2School of Medical Sciences, The University of Sydney, Camperdown, NSW 2006, Australia; 3Australian Infectious Disease Research Centre, School of Chemistry and Molecular Biosciences, University of Queensland, Brisbane, QLD 4072, Australia; n.west@uq.edu.au

**Keywords:** tuberculosis, vaccine, virulence, immune response, PdtaS

## Abstract

New control measures are urgently required to control tuberculosis (TB), as the current vaccine, Bacille Calmette–Guérin (BCG), has had a limited impact on disease spread. The identification of virulence mechanisms of *Mycobacterium tuberculosis* is an important strategy in vaccine design, as it permits the development of strains attenuated for growth that may have vaccine potential. In this report, we determined the role of the PdtaS response regulator in *M. tuberculosis* virulence and defined the vaccine potential of a *pdtaS*-deficient strain. Deletion of *pdtaS* (*Mtb*^ΔpdtaS^) resulted in reduced persistence of *M. tuberculosis* within mouse organs, which was equivalent to the persistence of the BCG vaccine in the lung and liver of infected mice. However, the generation of effector CD4^+^ and CD8^+^ T cells (CD44^+^CD62L^lo^KLRG1^+^) was similar between wild-type *M. tuberculosis* and *Mtb*^ΔpdtaS^ and greater than that elicited by BCG. Heightened immunity induced by *Mtb*^ΔpdtaS^ compared to BCG was also observed by analysis of antigen-specific IFN-γ-secreting T cell responses induced by vaccination. *Mtb*^ΔpdtaS^ displayed improved protection against aerosol *M. tuberculosis* compared to BCG, which was most apparent in the lung at 20 weeks post-infection. These results suggest that the deletion of the PdtaS response regulator warrants further appraisal as a tool to combat TB in humans.

## 1. Introduction

It is estimated that one-third of the world is infected with *Mycobacterium tuberculosis*, the bacterial agent responsible for tuberculosis (TB). More than 10 million new cases arise every year, with approximately 1.3 million deaths annually [1]. Although there is a vaccine available, *Mycobacterium bovis* Bacille Calmette–Guérin (BCG), the vaccine is highly variable in its protective efficacy [2] and has failed to control the TB epidemic, as new TB cases and deaths remain alarmingly high. Several novel vaccine candidates have entered clinical trials, including modified BCG, recombinant viral vector vaccines, protein in adjuvant, and attenuated *M. tuberculosis* (reviewed in [3]). Of these candidates, MTBVAC, deleted of the *phoP* and *fadD26* genes, was the first attenuated *M. tuberculosis* vaccine to reach clinical trials [4]. Unlike BCG, attenuated *M. tuberculosis* strains have the advantage of containing most genes present in *M. tuberculosis;* for example, BCG lacks region of difference 1 (RD1) of *M. tuberculosis* [5], which codes for several important antigens, including early secretory antigenic target protein 6 (ESAT6) and culture filtrate protein 10 (CFP10) [6].

Two-component systems (TCS) are used by most bacteria to respond rapidly to changing environments. This includes pH, temperature, osmolarity, and nutrient concentrations. Some systems of *M. tuberculosis* have been shown to regulate the expression of virulence factors, such as PhoP-PhoR and SenX3-RegX3 [7,8]. These regulatory protein pairs are composed of a membrane-bound histidine kinase sensor and a cytosolic, DNA-binding response regulator. Histidine kinase will auto-phosphorylate a conserved histidine residue after sensing a stimulus with the phosphoryl group and then transfer it to a conserved aspartate in the receiver domain of the response regulator. A response to the stimulus is enacted when the response regulator interacts with DNA binding motifs associated with promoter regions of target genes [9]. PhoP controls ~2% of *M. tuberculosis’s* coding capacity, and importantly, it controls the secretion of the virulence factor ESAT6 [10]. *M. tuberculosis* deleted of *phoP* displayed attenuated growth within mice, afforded protection against *M. tuberculosis* challenge in mice and induced long-term CD4^+^ T cell memory responses in mice that were not observed with BCG [8,11,12,13]. To create MTBVAC, a second attenuating deletion of fadD26 [14] was introduced to fulfill the Geneva Consensus requirements for the development of live TB vaccines [15]. MTBVAC has shown promising safety and immunity in early clinical trials and entered Phase III testing of efficacy in infants in 2022 [16].

The strongly attenuated phenotype of the *phoP* deletion mutant raises the possibility that other members of the TCS family may serve as targets of improved live vaccines. There are 12 two-component system pairs and 7 orphan genes in *M. tuberculosis*, which is low compared to other bacteria [9,17]. One of the least studied pairs is PdtaS/PdtaR (PdtaSR). Unlike canonical response regulators, PdtaR appears to act at the level of transcriptional antitermination rather than transcriptional initiation, hence the designation as a phosphorylation-dependent transcriptional antitermination regulator (PdtaR) [18]. Histidine kinase Rv3220, termed PdtaS, was shown to phosphorylate pdtaR but not other response regulators [19]. Based on structural data, PdtaS does not appear to be membrane-associated like other histidine kinases [19]. Orthologs of PdtaSR have been found in most Gram-positive bacteria Phyla, which also have a histidine kinase that is not membrane-anchored and a response regulator involved in transcriptional antitermination [20]. In *Mycobacterium smegmatis*, PdtaS appears to control susceptibility to antibiotics targeting ribosomes [21], and Cyclic di-GMP can bind to *M. smegmatis* PdtaS with high affinity and enhance its autokinase activity [22]. More recent findings have demonstrated that copper and nitric oxide regulate the PdtaS sensor kinase, with PdtaS/PdtaR forming part of a regulatory circuit that promotes bacterial virulence [23].

In this report, we utilized an *M. tuberculosis* transposon mutant of *pdtaS,* a member of a largely uncharacterized two-component regulator of *M. tuberculosis*, and found it to be attenuated for growth within host cells or lungs of mice. A *pdtaS*-deleted strain of *M. tuberculosis* was constructed and compared to virulent *M. tuberculosis* and the existing BCG vaccine for in vivo persistence, immunogenicity, and protective efficacy against aerosol *M. tuberculosis* at both acute and chronic stages of infection.

## 2. Materials and Methods

### 2.1. Mice

Female C57BL/6 mice, aged 6 weeks, were purchased from the Animal Resources Centre (Perth, Australia) or from Australian BioResources (Moss Vale, NSW, Australia) and were housed in the Centenary Institute mouse facility. All experiments were approved by the Sydney Local Health District Animal Welfare Committee (protocol 2020/009). For animal use, we adhered to the Australian Code for the Care and Use of Animals for Scientific Purposes 8th edition (2013).

### 2.2. Bacteria Culture Conditions

*M. tuberculosis* H37Rv, *M. tuberculosis* MT103, MT103:^ΔpdtaS^, *M. bovis* BCG Pasteur, and the transposon mutant strains (detailed below) were grown at 37 °C in 7H9 media (Difco Laboratories, BD Diagnostic Systems, Sparks, MD, USA) supplemented with 10% albumin-dextrose-catalase (ADC), 0.5% glycerol, and 0.05% Tween 80. The antibiotics hygromycin (50 μg/mL) or kanamycin (25 μg/mL) were added to liquid or solid media where required. 

### 2.3. Selection and Development of M. tuberculosis pdtaS Mutant Strains

*Mtb*-pdtaS^Tn^, *Mtb*-pdtaR^Tn^ and *Mtb*-mtrA^Tn^ were selected from a Tn*5* transposon mutant library [24]. To generate *Mtb:^ΔpdtaS^*, genomic DNA from *M. tuberculosis* H37Rv was used to amplify the upstream and downstream flanking regions of the *pdtaS* gene by PCR. The upstream region was inserted into the *StuI* and *XbaI* restriction sites, and the downstream region into the *HindIII* and *SpeI* restriction sites of the pYUB854 vector [25], placing the DNA fragments on either side of the hygromycin resistance gene, thus creating pYUB854-Δ*pdtaS*. To facilitate the deletion of *pdtaS*, *M. tuberculosis* MT103 containing the recombineering plasmid pJV53 [26] was used. The pJV53 protein contains recombination proteins, under the control of the inducible acetamidase promoter, that facilitate allelic exchange in mycobacteria [26]. Electrocompetent cells of MT103-pJV53 were prepared with the addition of acetamide (0.2%) for 16 h before 10% glycerol washes, as described previously [26]. The pYUB854-Δ*pdtaS* plasmid was linearized and 100 ng electroporated into MT103-pJV53 competent cells. After 72 h recovery in complete 7H9 media, electroporated cells were plated on 7H11 agar supplemented with 10% oleic acid–albumin–dextrose–catalase (OADC) and 0.5% glycerol (Difco Laboratories) containing hygromycin (100 mg/mL) and kanamycin (50 mg/mL). Deletion of the *pdtaS* gene from MT103 was confirmed by sequencing. 

### 2.4. In Vitro Infection of Macrophages

BCG, *M. tuberculosis* MT103, or the transposon mutant strains were added to 1 × 10^5^ RAW 264.7 cells (ATCC, TIB-71), a murine macrophage-like cell line, in 200 μL RPMI supplemented with 10% fetal bovine serum (FBS), at a multiplicity of infection of 1:1. Cell viability was determined using Trypan blue. The plates were incubated at 37 °C/5% CO_2_ for 4 h then washed gently 3 times with RPMI to remove extracellular bacteria. Fresh, complete RPMI was added to the cells, which were incubated for a further 7 days. At the 4 h or 7 day time points, the cells were lysed with dH_2_0/Triton-X-100 (0.01%). Colony forming units (CFU) were determined by growth at 37 °C on 7H11 agar. 

### 2.5. Vaccination and Infection of Mice

For vaccination, mice were injected subcutaneously (s.c.) with 5 × 10^5^ CFU of BCG, *Mtb*-pdtaS^Tn^ or *Mtb*:^ΔpdtaS^ (100 μL total volume). The route and dose were based on our previous studies in this model [13]. For infection, mice were anaesthetized by intraperitoneal (i.p.) injection of ketamine (100 mg/kg)/Xylazine (10 mg/kg) and were either intranasally administered with 50 μL of 10^3^ CFU *M. tuberculosis* strains or aerosol-infected via Middlebrook airborne infection apparatus with *M. tuberculosis* MT103 (Glas-Col, Terre Haute, IN, USA) to generate an initial infective dose of approximately 100 bacilli. Bacterial numbers were determined by homogenization of lungs, spleens, or livers after growth on 7H11 agar at 37 °C for approximately 3 weeks. 

### 2.6. Immunogenicity Studies

The lungs were perfused with PBS/heparin (20 U/mL), dissociated with collagenase I/DNase I in RPMI using a gentleMACS tissue dissociator (Miltenyi Biotec, Macquarie Park, NSW, Australia), and incubated for 30 min at 37 °C. The spleens and lungs were passed through 70 μm strainers, and red blood cells were lysed with ACK (ammonium–chloride–potassium) lysis buffer for 1 min before washing with RPMI. Cells were counted using a Countess Automated Cell Counter (Invitrogen, Waltham, MA, USA) before restimulation and antibody staining. FcγR receptors were blocked using anti-CD32/16, and cells were labeled with combinations of CD4-AlexaFluor 700, CD8-Pacific Blue (BD Biosciences, North Ryde, NSW, Australia), CD44-APC (eBioscience, San Diego, CA, USA), CD62L-PE, KLRG1-FITC (BioLegend, San Diego, CA, USA) surface antibodies, and live/dead UV blue dye (Life Technologies, Carlsbad, CA, USA). For intracellular cytokine staining, cells were prepared as above and restimulated overnight in the presence of 10 μg/mL *M. tuberculosis* culture filtrate protein (CFP; BEI Resources, Manassas, VA, USA) and 10 μg/mL Brefeldin A (Sigma-Aldrich, Castle Hill, NSW, Australia). The cells were surface stained and then permeabilized using Cytofix/Cytoperm^TM^ (BD Biosciences) and stained for intracellular IFN-γ-PE Cy7, TNF-APC and IL-2-PE (BD Biosciences). Samples were acquired on an LSR-Fortessa or LSR-II (Becton Dickinson) and analyzed using FlowJo^TM^ analysis software (Tree Star, Ashland, OR, USA, Version 9.7.6). Cells were gated by forward scatter height versus forward scatter area and side scatter height versus side scatter area to exclude doublets. A viability dye was included to gate out dead cells before the selection of relevant populations.

For IFN-γ ELISPOT, splenocytes and lung cells were restimulated overnight at 37 °C with 10 μg/mL *M. tuberculosis* culture filtrate protein (CFP) in wells pre-coated with α-IFN-γ (AN18). Cells were washed with PBS/Tween 20 (0.01%), and α-IFN-γ (XMG1.2-biotin) was added overnight at 4 °C. After washing, avidin alkaline phosphatase was added for 45 min at room temperature, then washed again, and alkaline phosphatase substrate solution was added (Sigma). The reaction was allowed to develop for 10 min before washing with water. The plates were analyzed using an AID ELISPOT reader and V6.0 software (AID, Strassberg, Germany).

### 2.7. Statistics

The significance of differences between experimental groups was evaluated by *t*-test or one-way analysis of variance (ANOVA), with a pairwise comparison of multi-grouped data sets achieved using Tukey’s post hoc test.

## 3. Results

### 3.1. Identifying In Vivo Attenuated M. tuberculosis Transposon Mutants of the Two-Component System Family

In order to identify *M. tuberculosis* genes with potential as vaccine candidates, an *M. tuberculosis* Tn5 transposon mutant library was utilized to identify genes that may influence growth after intranasal infection of mice. We validated the utility of the library by identifying a Tn5 mutant of the *mtrA* response regulator with a strongly attenuated phenotype when delivered to the mouse lung (Figure 1A) or spleen (Figure 1B). The residual growth of the Tn5 mutant differs from the full deletion of the gene, which is not viable [27] but does serve as an attenuation control for in vitro and in vivo studies. We also identified Tn5 mutants of both components of the *pdtaS/pdtaR* TCS (respectively). There was no significant decrease of *Mtb*-pdtaS^Tn^ and *Mtb*-pdtaR^Tn^ growth compared to wild-type *M. tuberculosis* in the mouse lung when assessed at the single time point of 4 weeks post-intranasal infection (Figure 1A,B). However, in vitro growth in macrophages revealed that *Mtb*-pdtaS^Tn^ grew at a slower rate than H37Rv (Figure 1C) and was more similar to the growth of BCG and *Mtb*-mtrA^Tn^ (Figure 1D). *Mtb*-pdtaR^Tn^ did not have a significantly decreased growth rate. In mice vaccinated with *Mtb*-pdtaS^Tn^ and then challenged 10 weeks later with aerosol *M. tuberculosis,* an approximate one log_10_ reduction in *M. tuberculosis* CFU in both the lung (Figure 1E) and spleen (Figure 1F) was observed compared to unvaccinated mice. This protection was significantly greater than that afforded by BCG in the lung and equivalent to the BCG-induced protection in the spleen (Figure 1). Therefore, a *pdtaS* Tn insertion mutant delivers an attenuated *M. tuberculosis* strain that can protect against pulmonary *M. tuberculosis* infection.

### 3.2. Persistence and Immunogenicity of a pdtaS Deletion Mutant of M. tuberculosis

Due to the attenuation of *Mtb*-pdtaS^Tn^ in macrophages and its protective efficacy against *M. tuberculosis* infection, a gene deletion mutant of *M. tuberculosis* was constructed to better define the vaccine potential of *M. tuberculosis* deficient in pdtaS (*Mtb*^ΔpdtaS^). Bacterial persistence was first defined in the organs of mice (spleen, liver, and lung) after vaccination. Wild-type *M. tuberculosis* (*Mtb*^WT^) increased over time in the lung (Figure 2A); conversely, bacterial load was reduced in the liver at later time points (Figure 2C). The highest *Mtb*^WT^ bacterial load was in the spleen, and the counts remained steady from day 10 to day 70 (Figure 2B). In contrast, recovery of *Mtb*^ΔpdtaS^ was low in organs at all timepoints, with bacteria only detected in the spleen at all timepoints examined (Figure 2B). BCG was not detected in the lung or liver after vaccination, and small numbers were recovered at only the day 30 timepoint in the spleen (Figure 2B). Therefore, the deletion of *pdtaS* results in an attenuated growth profile of *M. tuberculosis.*

To define the immunological profile after vaccination with *Mtb*^ΔpdtaS^, T cell immunity in vaccinated mice was examined. In both *Mtb*^ΔpdtaS^ and *Mtb*^WT^-vaccinated mice, a similar proportion of activated lung CD4^+^ T cells and CD8^+^ T cells was observed at day 10 post-infection (Figure 2D); however, these responses contracted over time (Figure 2E,F). The proportion of activated T cells in the lungs of BCG-vaccinated mice was similar to that observed in non-vaccinated animals at day 10, with BCG responses expanding by day 30 for both CD4^+^ and CD8^+^ T cells. Analysis of antigen-specific responses, by measuring the number of IFN-γ-secreting cells elicited after re-stimulation with *M. tuberculosis* culture filtrate protein (CFP), revealed rapid induction CFP-reactive, IFN-γ-secreting splenocytes at day 10 post-vaccination, with the magnitude of the response similar between *Mtb*^ΔpdtaS^ and *Mtb*^WT^ (Figure 3A). These responses contracted at longer timepoints (Figure 3B,C), although IFN-γ-secreting splenocytes were still detectable in all vaccinated groups at day 70 (Figure 3C). The number of lung cells secreting IFN-γ was near background levels for all groups at day 10 post-vaccination (Figure 3D), but high levels of responding cells were detected in the lung at day 30, with responses similar between all vaccinated groups (Figure 3E). On day 70, a similar high frequency of CFP-reactive IFN-γ-secreting was detected in BCG and *Mtb*^ΔpdtaS^-vaccinated groups, with the highest response observed in mice infected with *Mtb*^WT^ (Figure 3F). These data suggest that *Mtb*^ΔpdtaS^ induces rapid activation of T cells in mouse lungs after vaccination, with a corresponding influx of vaccine-specific cells secreting IFN-γ.

### 3.3. Protective Efficacy of Mtb^ΔpdtaS^ against Aerosol M. tuberculosis Infection

In order to determine if the heightened immunity pre-challenge correlated with protective efficacy induced by *Mtb*^ΔpdtaS^*,* mice were vaccinated s.c. with BCG or *Mtb*^ΔpdtaS^ and, after ten weeks, were challenged with low-dose (~100 CFU) aerosol *M. tuberculosis*. Both BCG or *Mtb*^ΔpdtaS^ significantly protected against *M. tuberculosis* infection in the lung at 4 weeks post-infection (Figure 4A). In the spleen, both vaccines reduced bacterial load compared to unvaccinated mice; however, a greater reduction was observed with *Mtb*^ΔpdtaS^ compared to BCG (Figure 4B). At week 20, BCG offered no protection in either the lung or spleen compared to unvaccinated mice (Figure 4C,D). However, *Mtb*^ΔpdtaS^ significantly reduced *M. tuberculosis* CFU in the lung at this timepoint (Figure 4C). Examination of CFP-reactive cytokine-producing T cells post-*M. tuberculosis* challenge revealed both BCG and *Mtb*^ΔpdtaS^ stimulation equivalent CD4^+^ T cell responses at week 4 post-challenge, which were most prominent for IFN-γ^+^TNF^+^ secreting cells, or cells expressing either IFN-γ^+^ or TNF alone (Figure 4E). While these phenotypes were also elevated at 20 weeks post-challenge, responses did not differ in unvaccinated mice that were challenged with *M. tuberculosis* (Figure 4F). These results reveal that *Mtb*^ΔpdtaS^ offers protection at both acute and chronic timepoints post-pulmonary *M. tuberculosis* infection, with protection correlating with the early expansion of cytokine-secreting CD4^+^ T cells in the lung. 

## 4. Discussion

A number of approaches have been used to develop improved vaccines for TB, including proteins in adjuvant, modified BCG or viral vectors expressing mycobacterial antigens [28]. An additional approach is to modify *M. tuberculosis* itself to attenuate the strain to a level safe for delivery to humans. Currently, one attenuated *M. tuberculosis* strain, MTBVAC, has progressed to a Phase III efficacy trial in infants [16]. MTBVAC contains a deletion of the two-component system protein PhoP, which mediates virulence and the generation of cellular immunity [8,13]. We hypothesized that the deletion of other members of this family may facilitate the development of attenuated *M. tuberculosis* vaccine strains with protective efficacy. Through analysis of a *pdtaS* transposon mutant and a gene-deficient strain, we identified *pdtaS* as a virulence determinant of *M. tuberculosis* with potential as a vaccine candidate.

The attenuated phenotype of *pdtaS*-deficient *M. tuberculosis* observed in this study suggests the PdtaSR TCS plays a role in *M. tuberculosis* virulence in vivo. This TCS pair has been shown to be structurally equivalent to the bacterial EutW/EutV system, which regulates ethanolamine catabolism; however, this does not appear to be the case in *M. tuberculosis* [20]. Rather, PdtaSR appears to play a role in the ability of *M. tuberculosis* to adapt to the toxic environment of the host cell. PdtaSR controls the regulation of genes involved in nitric oxide resistance, while purified PdtaS is regulated by zinc and copper [23], which may aid the coordination of the metal detoxification system for survival within host cells [29] and explain the reduced capacity of the *pdtaS* transposon mutant to replicate with macrophages (Figure 1). In addition, mycobacterial PdtaS is modulated by intracellular levels of cyclic di-GMP, providing a link between nutrient sensing and metabolic adaptation [22]. 

The attenuated phenotype of *M. tuberculosis* deficient in *pdtaS* mirrors the impact seen with deletion of other TCS, such as *phoPR*, *mtrAB*, *dosRST*, and *senXS/regS3* [17]. Conversely, disruption of other TCSs has resulted in an increased level of virulence. Deletion of *devR*, *trcS*, or *kdpDE* in *M. tuberculosis* resulted in a more rapid death after infection of SCID mice [30]. This suggests that *M. tuberculosis* may use TCS to balance its own survival and that of the host. Notably, the deletion of Rv3220c (*pdtaS*) in the study of Parish did not reveal a virulent phenotype. This may be a reflection of the different mouse strains or vaccination/infection routes used between studies. 

Both BCG and *Mtb*^ΔpdtaS^ demonstrated equivalent protection in the lung and spleen at an acute timepoint post-*M. tuberculosis* infection; however, in the chronic infection phase (20 weeks post-infection), BCG’s protection waned, while *Mtb*^ΔpdtaS^ maintained its ability to reduce *M. tuberculosis* CFU in the lung (Figure 4). The waning of BCG efficacy in animal models has been previously described [31] and may reflect the waning efficacy observed in human BCG efficacy trials [32]. We observed increased persistence of *Mtb*^ΔpdtaS^ in mice organs compared to BCG (Figure 2), and this may explain, in part, the enhanced protection of *Mtb*^ΔpdtaS^, as has been seen previously with other TB vaccine candidates [33,34]. Enhanced protection also correlated with earlier activation of T cells in the lung post-vaccination with *Mtb*^ΔpdtaS^ compared to BCG (Figure 2), as well as an increased frequency of vaccine-specific CD4^+^ T cells in the spleen after *Mtb*^ΔpdtaS^ delivery, suggesting early T cell priming (Figure 3). It is possible that the early expansion of T cells observed with the *pdtaS* vaccine compared to BCG results in a better ‘quality’ of T cells persisting at extended timepoints. It is worth noting that the generation of conventional CD4^+^ T cell responses does not always correlate with the protective efficacy of TB vaccine candidates [3], and other immune parameters not tested here may also contribute to the improved efficacy exhibited by *Mtb*^ΔpdtaS^.

## 5. Conclusions

This study identified that deletion of the *pdtaS* gene results in an attenuated strain of *M. tuberculosis* that induces robust expansion of antigen-specific T-cell responses and provides long-term protection in a murine model. The results presented here pave the way for further exploration and development of *Mtb*^ΔpdtaS^ as a live, attenuated TB vaccine, potentially in combination with additional mutations to improve safety and efficacy. However, a limitation of this study is that it only describes vaccine effectiveness in mice; thus, additional pre-clinical and clinical studies are warranted to ascertain the safety and efficacy of this vaccine candidate.

## Figures and Tables

**Figure 1 vaccines-12-00050-f001:**
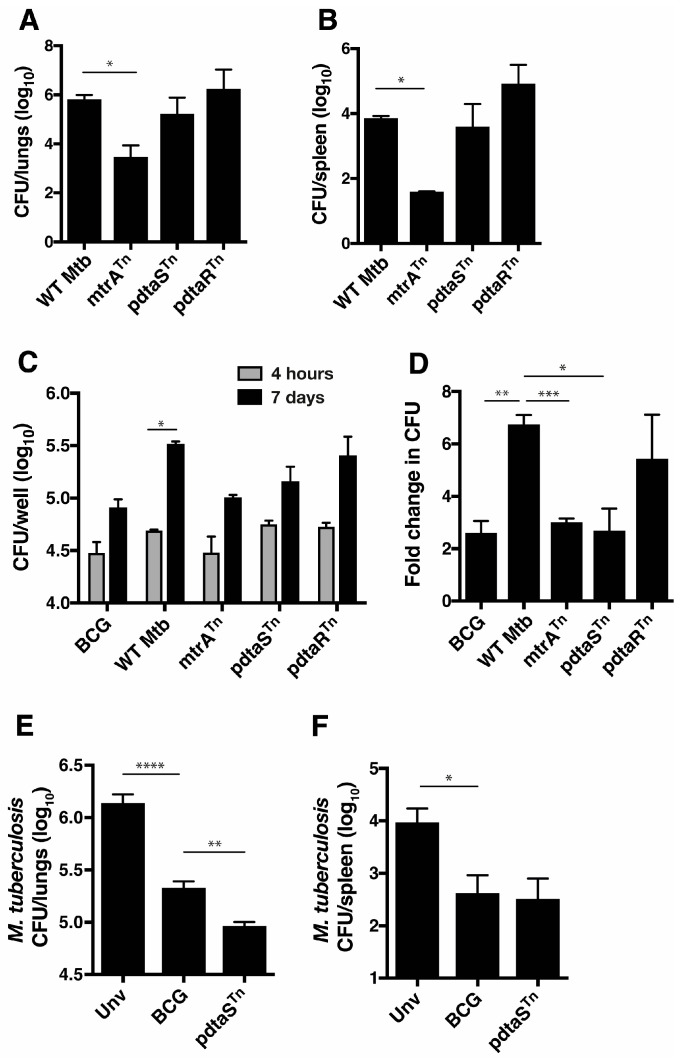
pdtaS^Tn^ growth is attenuated in macrophages and is protective as a vaccine against *M. tuberculosis* challenge. C57BL/6 mice (*n* = 4) were infected i.n. with 5 × 10^5^ CFU *M. tuberculosis* (WT *Mtb*), *Mtb*-mtrA^Tn^, *Mtb*-pdtaS^Tn^, or *Mtb*-pdtaR^Tn^. Four weeks after infection, the CFU in the lungs (**A**) and spleen (**B**) was determined. RAW 264.7 cells were infected with BCG, WT *Mtb*, *Mtb*-mtrA^Tn^, *Mtb*-pdtaS^Tn^, or *Mtb*-pdtaR^Tn^ (MOI 1:1) for 4 h or 7 days at 37 °C, the intracellular bacterial load was determined (**C**), and fold change in CFU was calculated (**D**). C57BL/6 mice (*n* = 5) were vaccinated s.c. with 5 × 10^5^ CFU BCG, *Mtb*-pdtaS^Tn^, or left unvaccinated. Ten weeks after vaccination, the mice were aerosol infected with *M. tuberculosis* H37Rv and 4 weeks after infection the *M. tuberculosis* CFU in the lungs (**E**) or spleens (**F**) was determined. The significances of differences between groups were determined by ANOVA (* *p* < 0.05, ** *p* < 0.01, *** *p* < 0.001, and **** *p* < 0.0001).

**Figure 2 vaccines-12-00050-f002:**
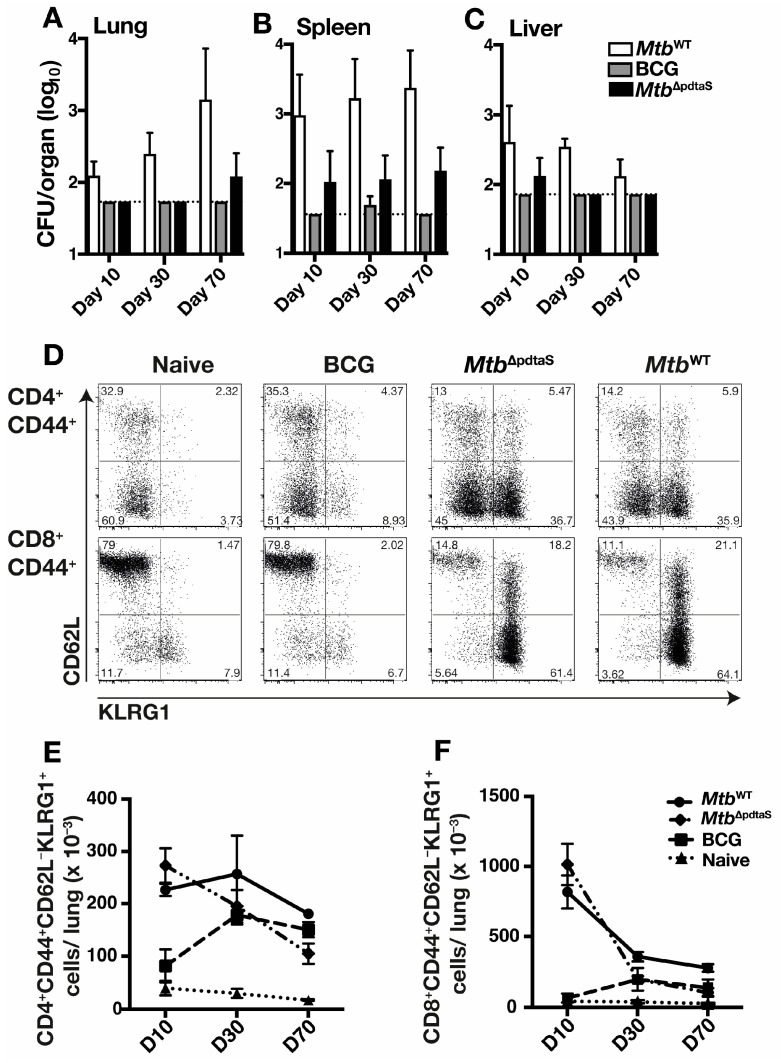
Immune activation and bacterial persistence of *Mtb*^ΔpdtaS^ in mice C57BL/6 mice (*n* = 4) were infected s.c. with 5 × 10^5^ CFU *M. tuberculosis* (*Mtb*^WT^), BCG, or *M. tuberculosis* deficient in *pdtaS* (*Mtb*^ΔpdtaS^). Ten, thirty or seventy days post-vaccination, bacterial CFU in the lungs (**A**), spleen (**B**), or liver (**C**) was determined. The dotted line is the limit of detection. Representative flow cytometry plots show day 10 activated T cells (CD62L^-^KLRG1^+^) after gating on CD4^+^CD44^+^ or CD8^+^CD44^+^ T cell populations (**D**). The total number of CD4^+^ (**E**) or CD8^+^ (**F**) activated T cells in the lungs is also shown. Results are representative of two independent experiments.

**Figure 3 vaccines-12-00050-f003:**
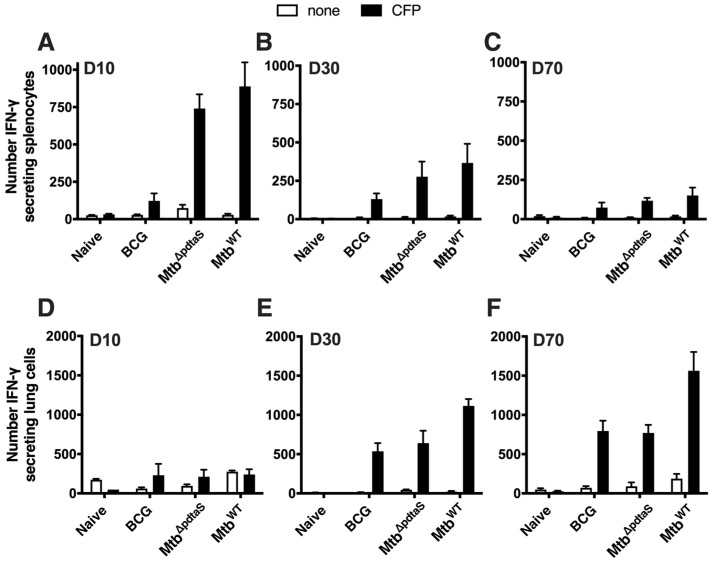
Antigen-specific IFN-γ production after vaccination with *Mtb*^ΔpdtaS^. C57BL/6 mice were vaccinated, as in Figure 2. Ten, thirty or seventy days post-vaccination, IFN-γ production was assessed by ELISPOT in the spleen (**A**–**C**) and lung (**D**–**F**) after restimulation with culture filtrate protein (CFP). Results are representative of two independent experiments.

**Figure 4 vaccines-12-00050-f004:**
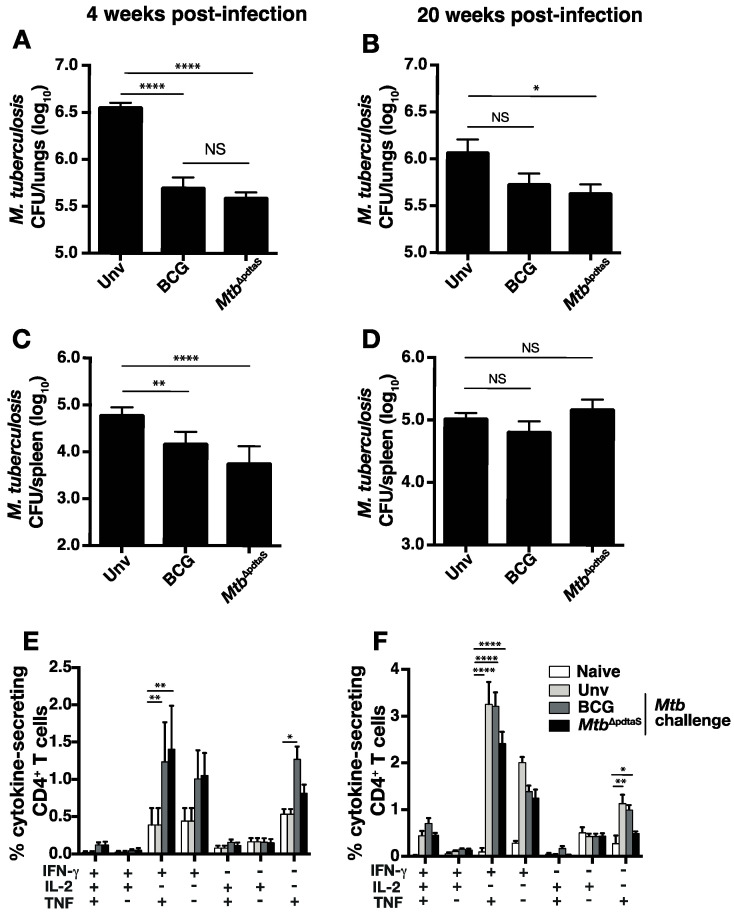
Protection afforded by Mtb^ΔpdtaS^ vaccination against *M. tuberculosis* aerosol challenge. C57BL/6 mice (*n* = 6) were vaccinated s.c. with 5 × 10^5^ CFU BCG or *Mtb*^ΔpdtaS^. Ten weeks after vaccination, the mice were aerosol-infected with ~100 CFU *M. tuberculosis*, and 4 (**A**,**C**) or 20 (**B**,**D**) weeks after infection, the *M. tuberculosis* load in the lungs (**A**,**B**) or spleen (**C**,**D**) was determined. The percentage of CD4^+^ lung T cells expressing intracellular IFN-γ, IL-2 or TNF after restimulation ex vivo with CFP at weeks 4 (**E**) and week 20 (**F**) post-challenge is also shown. Results are representative of two independent experiments. The significances of differences between groups were determined by ANOVA (* *p* < 0.05, ** *p* < 0.01, and **** *p* < 0.0001; NS = not significant).

## Data Availability

The data that support the findings of this study are available from the corresponding author upon reasonable request.

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
