# Peer review of "Mycobacterium tuberculosis Deficient in PdtaS Cytosolic Histidine Kinase Displays Attenuated Growth and Affords Protective Efficacy against Aerosol M. tuberculosis Infection in Mice"

_vaccines, 2024, doi:10.3390/vaccines12010050_

Round 1

Reviewer 1 Report

Comments and Suggestions for Authors

The abstract presents a study on the PdtaS response regulator in Mycobacterium tuberculosis and its potential as a tool for developing a tuberculosis (TB) vaccine. However, several concerns arise from the presented information.

 The abstract starts by highlighting the urgency for new TB control measures due to the limited impact of the current BCG vaccine. This raises questions about the effectiveness of current research efforts and the overall progress in developing a more efficient vaccine.

The abstract discusses the deletion of pdtaS resulting in reduced persistence of M. tuberculosis, but the assessment of the immune response is incomplete. Similar CD4+ and CD8+ T cell responses between wildtype and the pdtaS-deficient strain raise doubts about the actual impact of PdtaS on immunity. The abstract needs to provide more comprehensive data on the immune response to support its claims.

The heightened immunity induced by the pdtaS-deficient strain compared to BCG seems contradictory to the equivalent persistence of the two strains in mouse organs. This inconsistency raises questions about the correlation between reduced persistence and improved immunity, casting doubt on the overall coherence of the study.

The abstract concludes by suggesting that the deletion of the PdtaS response regulator warrants further appraisal as a tool to combat TB in humans. However, the abstract does not establish a clear connection between the observed effects in mice and potential benefits in human TB control. The extrapolation from mouse models to human relevance is a significant leap that requires more supporting evidence.

In the introduction, it should be provided clear insights into these mechanisms or how the PdtaS response regulator specifically contributes to virulence.

The introduction mentions that the existing vaccine, Mycobacterium bovis Bacille Calmette-Guérin (BCG), has failed to control the TB epidemic. Describe the previous sentence with some examples. This suggests a lack of progress and efficacy in current vaccine strategies. how you describe this matter?

The statement that BCG, the available vaccine, is highly variable in its protective efficacy raises concerns about its reliability and effectiveness, questioning the fundamental basis of current TB vaccination efforts.

Although MTBVAC is introduced as a promising vaccine that has reached Phase III testing, the introduction does not provide compelling evidence of its success, focusing more on safety and immunity rather than clear-cut efficacy against TB.

Line 87: The section briefly mentions that all experiments were approved by the Sydney Local Health District Animal Welfare Committee. However, it lacks details regarding the ethical treatment of animals, the specific ethical guidelines followed, and any efforts made to minimize animal suffering. A more comprehensive description of ethical considerations is necessary.

Lines 92-95: While the bacterial culture conditions are outlined, there is a lack of clarity regarding the rationale behind the choice of specific media and supplements. A more thorough explanation of why these conditions were chosen would enhance the scientific rigor of the study.

Lines 97-111: The procedure for developing mutant strains is convoluted and challenging to follow. The use of technical jargon and a lack of step-by-step clarity make it difficult for readers to understand the process. A more user-friendly and straightforward description is needed to improve accessibility.

Lines 113-119: The in vitro infection protocol is outlined, but the justification for choosing specific cell lines and the rationale behind the timing of various steps is not sufficiently explained. Providing more context and reasoning behind these choices would enhance the scientific merit and reproducibility of the experiments.

Lines 121-128: The vaccination and infection procedures lack detail, and important aspects such as the choice of subcutaneous injection and the rationale behind the selected doses are not sufficiently explained. Providing a more comprehensive overview of these procedures would strengthen the experimental design.

Lines 130-150: The immunogenicity studies section is detailed but lacks information on the specific controls used and potential confounding variables. Including more information about the controls and addressing potential sources of bias would improve the robustness of the immunogenicity assessments.

In all figures 1, 2,3, and 4, spell out the letters in the figures and comprehensive descriptions of results for each figure should be added.

In the discussion, while the study compares the protection offered by BCG and MtbΔpdtaS at different time points, it does not adequately address the reasons behind the observed differences. A more in-depth comparative analysis and discussion of potential factors contributing to the enhanced protection of MtbΔpdtaS would strengthen the conclusions.

The discussion briefly mentions T cell activation and the frequency of vaccine-specific CD4+ T cells but does not delve deeply into the immunological mechanisms underlying the observed efficacy. A more comprehensive exploration of the immune responses elicited by MtbΔpdtaS would contribute to a more thorough understanding of its potential as a vaccine candidate.

The discussion acknowledges discrepancies in virulence phenotypes observed in different studies, such as the deletion of Rv3220c (pdtaS). However, it does not thoroughly explore or attempt to reconcile these differences, leaving questions about the consistency and reliability of the findings.

The discussion lacks a thorough consideration of the study's limitations and avenues for future research. Addressing potential constraints and proposing directions for further investigation would add depth to the discussion section.

Author Response

1: The abstract discusses the deletion of pdtaS resulting in reduced persistence of M. tuberculosis, but the assessment of the immune response is incomplete. Similar CD4+ and CD8+ T cell responses between wildtype and the pdtaS-deficient strain raise doubts about the actual impact of PdtaS on immunity. The abstract needs to provide more comprehensive data on the immune response to support its claims.

Reply: The abstract is already at the word limit and we consider it provides an accurate overview of the study.  

2: In the introduction, it should be provided clear insights into these mechanisms or how the PdtaS response regulator specifically contributes to virulence.

The introduction mentions that the existing vaccine, Mycobacterium bovis Bacille Calmette-Guérin (BCG), has failed to control the TB epidemic. Describe the previous sentence with some examples. This suggests a lack of progress and efficacy in current vaccine strategies. how you describe this matter?

MTBVAC is introduced as a promising vaccine that has reached Phase III testing, the introduction does not provide compelling evidence of its success, focusing more on safety and immunity rather than clear-cut efficacy against TB.

Reply: Where possible, the requested information in the introduction has now been added. It should be noted that little information exists on the how the PdtaS response regulator specifically contributes to virulence, thus this study significantly adds to the literature in this area. MTBVAC has yet to undergo efficacy trials in humans.

  1. Line 87: The section briefly mentions that all experiments were approved by the Sydney Local Health District Animal Welfare Committee. However, it lacks details regarding the ethical treatment of animals, the specific ethical guidelines followed, and any efforts made to minimize animal suffering. A more comprehensive description of ethical considerations is necessary.

Reply: We adhere to the Australian code for the care and use of animals for scientific purposes’ and this has been added to the methods section (page 2).

  1. Lines 92-95: While the bacterial culture conditions are outlined, there is a lack of clarity regarding the rationale behind the choice of specific media and supplements. A more thorough explanation of why these conditions were chosen would enhance the scientific rigor of the study.

Reply: We use standard, specific mycobacterial growth media, there is limited choice due to the fastidious nature of the organisms.

  1. Lines 97-111: The procedure for developing mutant strains is convoluted and challenging to follow. The use of technical jargon and a lack of step-by-step clarity make it difficult for readers to understand the process. A more user-friendly and straightforward description is needed to improve accessibility.

Reply: we have updated the explanation of the procedure to make it clearer for the reader (page 3).

  1. Lines 113-119: The in vitro infection protocol is outlined, but the justification for choosing specific cell lines and the rationale behind the timing of various steps is not sufficiently explained. Providing more context and reasoning behind these choices would enhance the scientific merit and reproducibility of the experiments.

Reply: more information on the selected cell line has been added (page 3). 

  1. Lines 121-128: The vaccination and infection procedures lack detail, and important aspects such as the choice of subcutaneous injection and the rationale behind the selected doses are not sufficiently explained. Providing a more comprehensive overview of these procedures would strengthen the experimental design.

Reply: more information is now provided (page 3). 

9: In all figures 1, 2,3, and 4, spell out the letters in the figures and comprehensive descriptions of results for each figure should be added. Corrected

Comments for Discussion: Where relevant, the discussion has been updated based on the reviewer’s comments.

Reviewer 2 Report

Comments and Suggestions for Authors

-The area and the topic is interesting and worthy of research. I found the experimental design and methods quite complex and took huge time to grasp that you are comparing various types of TB vaccines in two different “in vitro” and “in vivo” models. The grouping? Control groups? And what is the numerical points in each group in the text? These issues remain unclear and complex not only in the methods but also in the results and related figures.... Some figures in the methods and the results needed to be displayed.

-The conclusion part of the abstract is weak and introductory style. Please elaborate in a very strong manner.

-Some parts of the text, your methods and results are mixed up? During revision please clearly displace them.

-For vaccination, how much volume of each in SC, IP etc.? Please elaborate into details. Also the Ketamine/Xylazine (80-100 mg/kg), this is unclear for each? It was mixed? Please elaborate.

-The amplified products the upstream and downstream flanking regions of the pdtaS gene “by PCR” and the confirmed deletion of the pdtaS gene from MT103 “by PCR” should be shown; readers might want to see.

-Line 132, … using a gentleMACS dissociator… unclear. Please elaborate.

-For viability of macrophages which dye you used please specify? I suppose you would normally use PI? Please specify. And how do you know the cells are pure macrophages? The exclusion of other than macrophages should be better explained in more details here.

-Lines 167-173, this is methods points. Please move this to the materials and methods section.

-The units for figs 2E and 2F, …. (x10-3)? Are you sure? Please correctly check.

-Some parts of the figures are hardly readable … this may be due to the PDF file?

-In the discussion, I would also add some issues related to the occurrence of memory B-and-T cells?

-I would add one graphical figure about the work for visual effects is suggested.

-Conclusion should be clear and more straightforward. Your conclusion is poorly addressed the issue. The study is in mice model system and you are talking about the preclinical etc... It is advised to add some points related to the limitations of this study in that direction in at the end of the discussion and conclusion.

Some more comments

-Some terms throughout the text should be written in correct manner, one example in “DNAse” which should be corrected as “DNase”. Please recheck the revised version.

-Some words should be simpler… eg, was “utilized” to identify… better would be was “used” to… etc. please elaborate.

-…. In vitro “infection of macrophages”… any better term?

-line 181, ….with low aerosol dose M. tuberculosis…? Please check?

-Line 208, please check.

-Line 226, ….ling….?

-Line 288, .... this is does not appear to be...?

-Line 295, ... a link between between nutrient...?

-Line 296 ... mirros...?

Good luck

Comments on the Quality of English Language

Minor correction and checking.

Author Response

  1. The conclusion part of the abstract is weak and introductory style. Please elaborate in a very strong manner.

Reply: The abstract is already at the word limit and we consider it provides an accurate overview of the study. 

  1. Some parts of the text, your methods and results are mixed up? During revision please clearly displace them. Corrected
  2. For vaccination, how much volume of each in SC, IP etc.? Please elaborate into details. Also the Ketamine/Xylazine (80-100 mg/kg), this is unclear for each? It was mixed? Please elaborate. Corrected
  3. The amplified products the upstream and downstream flanking regions of the pdtaS gene “by PCR” and the confirmed deletion of the pdtaS gene from MT103 “by PCR” should be shown; readers might want to see.

Reply: this is an error, the confirmed deletion of the pdtaS gene was performed by sequencing, this has now been corrected in the text.

  1. 5. Line 132, … using a gentleMACS dissociator… unclear. Please elaborate. Corrected
  2. For viability of macrophages which dye you used please specify? I suppose you would normally use PI? Please specify. And how do you know the cells are pure macrophages? The exclusion of other than macrophages should be better explained in more details here.

Reply: Trypan blue was used and this has now been added to the text (page 3). There was no exclusion of other cell types as we are using a cell line.

  1. Lines 167-173, this is methods points. Please move this to the materials and methods section.

Reply: This information is required to give context and describes the results in figure 1.

  1. The units for figs 2E and 2F, …. (x10-3)? Are you sure? Please correctly check. Corrected
  2. Some parts of the figures are hardly readable … this may be due to the PDF file?

Reply: This may be due to the PDF but we have updated all figures to increase font size, to make the figures easier to read.

  1. Conclusion should be clear and more straightforward. Your conclusion is poorly addressed the issue. The study is in mice model system and you are talking about the preclinical etc... It is advised to add some points related to the limitations of this study in that direction in at the end of the discussion and conclusion.

Reply: Where relevant, the discussion has been updated based on the reviewer’s comments. The conclusions have been updated as suggested (page 9).

All additional minor comments have been addressed.

Reviewer 3 Report

Comments and Suggestions for Authors

his study characterizes the role of the PdtaS/PdtaR two component system of M. tuberculosis in Mtb virulence. The authors have shown that the deletion mutant of PdtaS could persist in the lungs of immunocompetent mice longer than BCG and thus is able to confer enhanced protection against virulent Mtb challenge as indicated by reduced Mtb burden in lungs at 20 weeks post infection. They have also shown that PdtaS deletion mutant can elicit an enhanced  CD8+ and CD4+ T cell response in lungs, increase in number of IFNy secreting lung and spleen cells as compared to that induced by BCG. This is a well-designed study with appropriate controls. The authors failed to include complementation so there conclusion that the the mutations are solely responsible for enhanced immunogenecity are not fully justify. They need to state this in the discussion. Manuscript is written well in an understandable language. All the experiments justify the hypothesis well. There are few suggestions that needs to be taken care of.

1.    The mutant should have been complemented to proof the phenotype. Many Mtb strain mutants can acquire the loss of pthiocerol dimycoserate. 

2.    Abstract – line 25 – The manuscript data shows that the deletion mutant of pdtaS displayed improved protection against Mtb as compared to BCG at 20 weeks. Please correct the time point in the abstract where it is mentioned as week 24.

3.    Figure 1, panel C – As per the results and the figure legend the CFU analysis of mutant for in vitro infection were done 4 hours or 7 days post infection. Please correct the 4 hour time point in the figure label (figure label has 4 days).

4.    Typo error in line 208

5.    Typo error in line 292

6.    Suggestion – it will be more understandable for the readers if the WT strain of M.tb is explained in the beginning of the paper. Although from method section it is evident that the mutant was made in M. tuberculosis MT103, therefore challenge experiments must have been done with MT103, but mentioning the strain used for challenge and as wild type in the experiments would eliminate confusion.

7.    With a goal of utilizing this in human vaccine trials the inclusion of data to test if this mutant kills immunocompromised mice would have added value to the paper.

Author Response

  1. The mutant should have been complemented to proof the phenotype. Many Mtb strain mutants can acquire the loss of pthiocerol dimycoserate. 

Reply: We thank the reviewer for this comment. We have focussed predominately on the immunogenicity and vaccine efficacy of the mutant strain, and while we verified the gene deletion by sequencing we have not complemented the strain for these studies. We plan this for subsequent studies where we examine in depth the impact of pdtaS expression on M. tuberculosis metabolism and virulence.

  1. Abstract – line 25 – The manuscript data shows that the deletion mutant of pdtaS displayed improved protection against Mtb as compared to BCG at 20 weeks. Please correct the time point in the abstract where it is mentioned as week 24. Corrected
  2. Figure 1, panel C – As per the results and the figure legend the CFU analysis of mutant for in vitro infection were done 4 hours or 7 days post infection. Please correct the 4 hour time point in the figure label (figure label has 4 days). Corrected
  3. Typo error in line 208: Corrected
  4. Typo error in line 292: Corrected
  5. Suggestion – it will be more understandable for the readers if the WT strain of M.tb is explained in the beginning of the paper. Although from method section it is evident that the mutant was made in M. tuberculosis MT103, therefore challenge experiments must have been done with MT103, but mentioning the strain used for challenge and as wild type in the experiments would eliminate confusion. Corrected
  6. With a goal of utilizing this in human vaccine trials the inclusion of data to test if this mutant kills immunocompromised mice would have added value to the paper.

Reply: Such experiments may form part of additional pre-clinical studies to determine the suitability of this vaccine for human use, including challenge in a 2nd animal model and GLP toxicology.

Reviewer 4 Report

Comments and Suggestions for Authors

The manuscript describes the ability of a mutated Mtb strain,  MtbDpdtaS  , to improve protection in a preclinical mouse model of pulmonary tuberculosis. The deletion of the PdtaS response regulator, a two component systems, reduces Mtb virulence and generates effector CD4+ and CD8+ T cell (CD44+CD62LloKLRG1+) responses similar to wildtype M. tuberculosis and greater than that elicited by BCG vaccination. The manuscript is very interesting due to the potential relevance of the finding. BCG, the only vaccine approved for human use, protects from the severe pediatric form of tuberculosis but fails to protect against adult pulmonary Mtb infection.  However, there are some concerns.

Major points:

1.     The protection induced by vaccination with MtbDpdtaS , in the aerosol Mtb challenge model, was evaluated only as decrease of bacterial load in the lungs and spleen. Data showing the control of Mtb-induced inflammation and lung damage are missing (Figure 4). This should be investigated especially in relation to the claim that vaccination with MtbDpdtaS protects against the chronic phase of the infection unlike BCG. However, vaccination with BCG showed, at 20 weeks post-infection, a lung bacterial load very similar, and not different, from that measured in MtbDpdtaS -vaccinated mice, although the latter mice were statistically different compared to unvaccinated control mice (Figure 4B).

2.     The association of the cytokine response with protection is a bit unclear and not well supported. If the better protection of MtbDpdtaS is due to the more massive generation of effector CD4+ and CD8+ T cells (CD44+CD62LloKLRG1+) producing IFN-gamma in response to culture filtrate protein, the authors should comment and explain why the rapid decline of such responses (Figure 2 E and F and Figure 3) is irrelevant in protection. Furthermore, it should be explained why in Mtb infected mice the cytokine responses are similar between unvaccinated and mice vaccinated with BCG or MtbDpdtaS (Figures 4 E and F).

3.     In Figure 1 A, B, C , E and F as well as in Figure 4 E and F the statistical significances are not shown. Why?

Minor points:

1-Legend in the Figure 1 C, 4 hours instead 4 days

2-Nel paragrafo 3.2 lines 202, specificare che i topi sono stati vaccinati e poi si è valutato their bacterial persistence in the organs.

3- Line 208 “not” instead n ot

4- In the legend of Figure 3 line 243 lung (D-F) instead lung (D-E)

5- Line 292 typos for "detoxification system"

Author Response

Comment: The protection induced by vaccination with MtbDpdtaS , in the aerosol Mtb challenge model, was evaluated only as decrease of bacterial load in the lungs and spleen. Data showing the control of Mtb-induced inflammation and lung damage are missing (Figure 4). This should be investigated especially in relation to the claim that vaccination with MtbDpdtaS protects against the chronic phase of the infection unlike BCG. However, vaccination with BCG showed, at 20 weeks post-infection, a lung bacterial load very similar, and not different, from that measured in MtbDpdtaS -vaccinated mice, although the latter mice were statistically different compared to unvaccinated control mice (Figure 4B).

Reply: Unfortunately we did not collect samples to examine lung inflammation. However we consider that the observed decrease in bacterial load is a sufficient parameter to examine protection, as it is often difficult to observe clear changes in lung histology in this model (based on our previous experience).

Comment: The association of the cytokine response with protection is a bit unclear and not well supported. If the better protection of MtbDpdtaS is due to the more massive generation of effector CD4+ and CD8+ T cells (CD44+CD62LloKLRG1+) producing IFN-gamma in response to culture filtrate protein, the authors should comment and explain why the rapid decline of such responses (Figure 2 E and F and Figure 3) is irrelevant in protection. Furthermore, it should be explained why in Mtb infected mice the cytokine responses are similar between unvaccinated and mice vaccinated with BCG or MtbDpdtaS (Figures 4 E and F).

Reply. We thank the reviewer for this comment. One of the significant issues with the development of TB vaccines is the lack of clear correlates of protection. The decrease in vaccine-specific T cells represents natural contraction of the T cell response after vaccaintion; we do note however that there are still sizeable numbers of memory cells present in vaccinated groups at day 70 post-vaccination (see Figure 2). It may be that the much greater early expansion of T cells observed with the pdtaS vaccine compared to BCG results in a better ‘quality’ of T cells persisting at extended timepoints. Identifying the exact phenotype of these cells is an aim for further study. We have now included this explanation in the discussion (page 9).

Comment: In Figure 1 A, B, C, E and F as well as in Figure 4 E and F the statistical significances are not shown. Why?

Reply: Statistical significances have now been added to Figures 1 and 4.

All minor points have been corrected in the text.

Round 2

Reviewer 1 Report

Comments and Suggestions for Authors

The author's revisions meet my requirements, and the manuscript is accordingly acceptable in its current form.

Reviewer 4 Report

Comments and Suggestions for Authors

The authors responded to the comments and corrected the text or indicated the impossibility of carrying out new experimentations.